# Genome-Wide Identification and Characterization of Oil-Body-Membrane Proteins in Polyploid Crop *Brassica napus*

**DOI:** 10.3390/plants11172241

**Published:** 2022-08-29

**Authors:** Wei Zhao, Jun Liu, Lunwen Qian, Mei Guan, Chunyun Guan

**Affiliations:** 1College of Agronomy, Hunan Agricultural University, Changsha 410128, China; 2Key Laboratory of Biology and Genetic Improvement of Oil Crops, Ministry of Agriculture and Rural Affairs, Oil Crops Research Institute, Chinese Academy of Agricultural Sciences, Wuhan 430062, China; 3Oil Crops Research, Hunan Agricultural University, Changsha 410128, China; 4Hunan Branch of National Oilseed Crops Improvement Center, Changsha 410128, China

**Keywords:** *Brassica napus*, oil body, lipid droplet, oleosin, caleosin, steroleosin, evolution

## Abstract

Oil-body-membrane proteins (OBMPs) are essential structural molecules of oil bodies and also versatile metabolic enzymes involved in multiple cellular processes such as lipid metabolism, hormone signaling and stress responses. However, the global landscape for *OBMP* genes in oil crops is still lacking. Here, we performed genome-wide identification and characterization of *OBMP* genes in polyploid crop *Brassica napus*. *B. napus* contains up to 88 *BnaOBMP* genes including 53 oleosins, 20 caleosins and 15 steroleosins. Both whole-genome and tandem duplications have contributed to the expansion of the *BnaOBMP* gene family. These *BnaOBMP* genes have extensive sequence polymorphisms, and some harbor strong selection signatures. Various cis-acting regulatory elements involved in plant growth, phytohormones and abiotic and biotic stress responses are detected in their promoters. *BnaOBMPs* exhibit differential expression at various developmental stages from diverse tissues. Importantly, some *BnaOBMP* genes display spatiotemporal patterns of seed-specific expression, which could be orchestrated by transcriptional factors such as EEL, GATA3, HAT2, SMZ, DOF5.6 and APL. Altogether, our data lay the foundations for studying the regulatory mechanism of the seed oil storage process and provide candidate genes and alleles for the genetic improvement and breeding of rapeseed with high seed oil content.

## 1. Introduction

*Brassica napus* L. (known as rapeseed, oilseed rape and rapa), a relatively recent allotetraploid formed from hybridization between *Brassica rapa* and *Brassica oleracea*, is an important oilseed crop and widely cultivated in the world [1,2]. Rapeseed provides more than 13% of the world’s supply of vegetable oil [3]. Thus, increasing seed oil content (SOC) is one of the most important breeding objectives for rapeseed with the greatest economic significance [4]. Over the past decades, many QTLs (quantitative trait loci) controlling SOC have been identified in *B. napus*, and significant progress has been achieved in breeding rapeseed varieties with high oil content [5,6]. However, the molecular mechanism underlying the regulation of seed oil metabolism, particularly the storage process, is still poorly understood in *B. napus*.

Lipids, mainly triacylglycerols (TAGs), are the main reserves in *B. napus* seeds and stored in oil bodies (OBs; known as lipid droplets, oil droplets or oleosomes) [7]. OBs are specialized organelles acting as neutral lipid storage compartments, which are widespread in both prokaryotic and eukaryotic cells [8,9,10,11]. The formation of OBs begins on the endoplasmic reticulum (ER), where TAGs are synthesized in maturing seeds, and then bud off from the ER membrane as droplets consisting of an oil core and a hydrophilic surface with a phospholipid monolayer [12]. The structure and size of OBs are highly dynamic during seed development, and various oil-body-membrane proteins (OBMPs) wrap around the surface to stabilize the OBs and exert metabolic functions [13,14,15]. Previous studies showed that overexpression of some *OBMP* genes could regulate oil body size, as well as seed size and weight, to influence oil accumulation in seeds [16,17,18].

The *OBMP* gene family is composed of three main classes including oleosin, caleosin and steroleosin [19]. Oleosin is a relatively small protein of 15~26 kilodaltons (kDa), with a conserved central domain of approximately 70 uninterrupted nonpolar residues, which could form a hydrophobic hairpin [20,21]. The two arms of this hairpin are linked by a proline (Pro) loop (called Pro knot) with three conserved Pro residues and one Ser residue. The N- and C-terminal peptides of oleosin could form amphipathic α-helical structures interacting horizontally with the phospholipid monolayer, acting as a receptor for the binding of metabolic enzymes or regulatory proteins. Oleosin genes are widespread from green algae to higher plants [20]. Six oleosin lineages have been recognized including primitive (P), universal (U), tapetum (T), mesocarp (M), and seed low-molecular-weight (SL) and high-molecular-weight (SH) oleosins [22,23,24]. The *Arabidopsis thaliana* genome has 17 oleosin genes of which *OLE1* and *OLE2* are the most abundant in seeds. Double oleosin mutant (i.e., *ole1* and *ole2*) seeds have enlarged OBs and exhibit severely defective germination [25]. In addition, the seeds of the oleosin mutants are sensitive to freezing stress [26], indicating that oleosins function in enhancing plant survival during winter.

Like oleosins, caleosins are also structural proteins of OBs, consisting of a conserved hydrophobic central domain with a proline knot motif. Particularly, they contain an N-terminal hydrophilic domain with a single Ca^2+^-binding site, EF-hand motif, and possess an enzyme activity as peroxygenase [27]. Caleosin genes have been found ubiquitously in plants [28]. Among the eight caleosins identified in *A. thaliana*, *CLO1* is preferentially expressed in developing seeds, and *clo1* mutant seeds exhibit distorted vacuole morphology and a significant delay in the storage lipid degradation [29]. In addition, caleosins display calcium-binding properties and play crucial roles in many aspects of growth and development such as cell division, photosynthesis, polarity formation, apoptosis and stress resistance [30,31]. Steroleosins are known as sterol dehydrogenases due to the fact of their sterol-binding and sterol-coupling dehydrogenase activity [9,32,33], and they are reported to have functions related to sterol-regulated signal transduction, seed maturation and germination [33]. Unlike oleosin and caleosin, steroleosin has two main structural domains including an N-terminal hydrophobic domain, which contains a conserved proline knob instead of the proline knot, and a C-terminal domain exhibiting an NADP(H)-binding subdomain and a sterol-binding subdomain [9,32]. Various steroleosin genes have been discovered in *A. thaliana* [34], *Pinus massoniana* [35] and *Sesamum indicum* [36].

Although *OBMP* family genes have been studied in several plant species, the comprehensive identification and characterization of this gene family from important oil seed corps is lacking. In the present study, we performed a genome-wide identification of *OBMP* family genes in *B. napus* and 54 other genomes from green algae to angiosperms. The gene structures, phylogenetic relationships, cis-acting regulatory elements in promoters, sequence polymorphisms and expression patterns of *BnaOBMPs* were analyzed. We also identified transcriptional factors that potentially regulate the expression of *BnaOBMP* genes during seed development. Our results help to further elucidate the molecular mechanism of the seed oil storage process and provide new targets for the selection of rapeseed with high SOC.

## 2. Results

### 2.1. Identification, Properties and Genomic Distributions of BnaOBMP Genes

To identify all the members of the *OBMP* gene family in rapeseed, we used protein sequences of 33 *A. thaliana OBMP* genes, including 17 oleosins, 8 caleosins and 8 steroleosins (Appendix A), as queries to search against the protein dataset of *B. napus* var. ZS11 with BLASTP setting the E-value at 1 × 10^−5^. The peptides of putative *BnaOBMPs* with the best hits on *A. thaliana* oleosin, caleosin or steroleosin proteins were further used to predict functional domains in the Pfam database to confirm the presence of the oleosin domain (PF01277), the caleosin domain (PF05042) or the steroleosin domain (PF00106). Finally, a total of 88 *BnaOBMP* genes were identified in rapeseed of which 43 and 45 genes originated from the A and C subgenomes, respectively (Table 1). Based on the functional domains of each *BnaOBMP* contained, these genes could be divided into 53 oleosins, 20 caleosins and 15 steroleosins (Table 1). The 53 oleosin genes could be further classified into four subfamilies of 27 T, 8 SL, 9 SH and 9 U oleosins in rapeseed. Meanwhile, we also found 44 and 47 *OBMP* genes in its two ancestors of *B. rapa* var. Z1 and *B. oleracea* var. HDEM, respectively (Appendix A). Additionally, we identified *OBMP* genes in fifty-one other genomes covering green algae, mosses, gymnosperms and angiosperms and found that *B. napus* contained the greatest number of *OBMP* family genes (Appendix A).

The transcript length of *BnaOBMPs* varied from 240 (*BnaOBMP.T6*) to 1392 (*BnaOBMP.S13*) base pairs (bp). All identified *BnaOBMP* genes encoded proteins with a size ranging from 79 (*BnaOBMP.T6*) to 463 (*BnaOBMP.S13*) amino acids (aa), a molecular weight (MW) from 7.93 (*BnaOBMP.T6*) to 51.97 (*BnaOBMP.S13*) kDa, an isoelectric point (pI) from 4.93 (*BnaOBMP.C4*) to 11.56 (*BnaOBMP.T3*), an instability index from 13.84 (*BnaOBMP.SH5*) to 66.84 (*BnaOBMP.S13*), an aliphatic index from 48.8 (*BnaOBMP.T25*) to 139.32 (*BnaOBMP.T20*) and a grand average of hydropathy (GRAVY) from −0.864 (*BnaOBMP.T13*) to 1.597 (*BnaOBMP.T6*) (Table 1). Twenty-six BnaOBMPs were predicted to be localized in the nucleus of which six were also predicted to be mitochondria-localized proteins (Table 1).

To attain a holistic view of *BnaOBMP* gene distribution on the rapeseed genome, the 88 *BnaOBMPs* were mapped on the corresponding chromosomes according to their physical positions. Almost all of the *BnaOBMPs* were localized in the chromosome regions with relatively higher gene density in rapeseed, particularly towards the terminal regions (Figure 1). The rapeseed genome had a very unequal distribution of *BnaOBMP* genes among the nineteen different chromosomes, and the gene numbers varied from one (C05 and A09) to eleven (A03). Moreover, some homologous chromosomes of the A and C subgenomes contained a comparable number of *BnaOBMP* genes such as four in both A01 and C01, eight in both A02 and C02, and eight in A10 and nine in C09. Notably, the T oleosin genes showed an aggregated distribution on the terminal regions of A02 (*BnaOBMP**.T1*, *BnaOBMP**.T2* and *BnaOBMP**.T3*), A03 (*BnaOBMP**.T4*, *BnaOBMP**.T5*, *BnaOBMP**.T6* and *BnaOBMP**.T7*), A10 (*BnaOBMP**.T10*, *BnaOBMP**.T11*, *BnaOBMP**.T12*, *BnaOBMP**.T13* and *BnaOBMP**.T14*), C02 (*BnaOBMP**.T15*, *BnaOBMP**.T16* and *BnaOBMP**.T17*), C03 (*BnaOBMP**.T18*, *BnaOBMP**.T19* and *BnaOBMP**.T20*) and C09 (*BnaOBMP**.T23*, *BnaOBMP**.T24*, *BnaOBMP**.T25*, *BnaOBMP**.T26* and *BnaOBMP**.T27*).

### 2.2. Phylogeny, Gene Structures and Crucial Motifs of BnaOBMPs

To explore the molecular evolution of the *OBMP* gene family in *B. napus*, a total of 212 *OBMP* genes from *B. napus*, *B. rapa*, *B. oleracea* and *A. thaliana* were used to construct an unrooted phylogenetic tree using the maximum likelihood (ML) method. According to the phylogenetic relationships of these *OBMP* genes, they could be divided into three independent classes, which is consistent with classification by the functional domains (i.e., oleosin, caleosin and steroleosin) contained (Figure 2a). Moreover, the oleosin class could be further categorized into four subclasses, corresponding to the SH, SL, T and U subfamilies (Figure 2a).

Based on the gene information of the genome available in the BnPIR database (http://cbi.hzau.edu.cn/bnapus/ (accessed on 8 January 2022), we performed a gene structure analysis on the *BnaOBMPs*. The adjacent *BnaOBMP* genes in the phylogenetic tree, which were derived from the homologous A and C subgenomes of rapeseed, respectively, exhibited similar gene structures (Figure 2b). For the *BnaOBMP* genes of the caleosin or steroleosin subfamilies, their exon features of order, length and number were largely conserved among different members (Figure 2b). On the contrary, the *BnaOBMP* genes from the oleosin subclasses showed varied gene structures. For examples, the SH and SL oleosins and most of the T oleosins contained two exons and one intron, while all of the U oleosins and three T oleosin genes (i.e., *BnaOBMP.T6*, *BnaOBMP.T12* and *BnaOBMP.T25*) had only one exon, and the *BnaOBMP.T8* and *BnaOBMP.T21* oleosin genes had four and three exons, respectively (Figure 2b). Additionally, the conserved motifs of BnaOBMP proteins were identified using MEME, and three distinct conserved motifs (i.e., Motifs 1, 2 and 3) were found (Figure 2b). Motif 1 was a proline knot with four invariable residues of the proline knot sequence (-PX_5_SPX_3_P-), Motif 2 was a Ca^2+^-binding site, and Motif 3 was an NADPH-binding domain (Figure 2c). The sequence homology of BnaOBMP proteins was determined through multiple sequence alignment. In *B. napus*, the protein sequences of caleosins consisted of an H-form insertion, a Ca^2+^-binding motif and a proline knot (Appendix A). All the oleosins comprised an N-arm of a Pro loop, a proline knot and a C-arm of a Pro loop (Appendix A), and the steroleosins harbored one proline knob, one NADPH-binding motif and a sterol-binding domain in the C-terminal (Appendix A).

### 2.3. Synteny and Gene Duplication Modes of BnaOBMP Genes

Gene duplications are considered to be one of the major driving forces in the evolution of genomes and expansions of gene families [37]. Whole-genome duplication, segmental duplication and tandem duplication are the major causes of gene family expansion in plants [38]. Based on the synteny and collinearity of *B. napus* genome from all-against-all pairwise alignment of all the proteins, we identified the duplicated events for *BnaOBMP* genes. Approximately 82.9% (73/88) of the *BnaOBMP* genes were found associated with at least one collinear gene pair (Figure 3a). A total of 165 gene pairs with 72 (accounting for 81.82%) genes were identified as WGD/segmental duplication (Figure 3a,b). In addition, 15 (accounting for 17.04%) tandem duplicated genes were identified within 6 tandem duplicated gene clusters, and they all belonged to the T oleosin subfamily (Figure 3a,b). To better understand different selective constraints on the *BnaOBMP* gene family, the nonsynonymous substitution/synonymous substitution (*Ka*/*Ks*) ratios of the *OBMP* gene pairs between *B. napus* and *A. thaliana* were calculated (Appendix A). Except for *BnaOBMP.T5*, *BnaOBMP.T10*, *BnaOBMP.T15*, *BnaOBMP.T18*, *BnaOBMP.T24*, *BnaOBMP.SH9*, *BnaOBMP.SL7* and *BnaOBMP.S13*, all other orthologous *OBMP* gene pairs had *Ka*/*Ks*  <  1. In addition, the *Ka*/*Ks* ratios of T oleosin genes were substantially higher than the other *OBMP* genes (Appendix A).

### 2.4. Cis-Acting Regulatory Elements in the Promoters of BnaOBMP Genes

The cis-acting elements in the promoter region play vital roles in the regulation of gene expression and also function to coordinate responses to developmental and environmental cues in plants [39]. To identify the putative cis-acting elements of *BnaOBMP* genes, the genomic DNA sequences from the transcription initiation site (+1) to 2 kilo bases (Kb) upstream of the *BnaOBMP* genes were extracted for cis-acting elements profiling using PlantCARE. All the *BnaOBMP* gene promoters possessed various cis-acting regulatory elements (Figure 4). Based on the functional annotation of these cis-acting elements, they could be classed into three main groups including plant growth and development, phytohormone responsive and abiotic and biotic stress responsive. Specifically, the *BnaOBMP* genes contained multiple phytohormone responsive elements, such as ABRE (abscisic acid-responsive element), AuxRE (auxin-responsive element), ERE (ethylene-responsive element), GARE (gibberellin-responsive element), MeJARE (MeJA-responsive element) and SARE (salicylic acid-responsive element), which suggests that the expression of *BnaOBMP* genes might be induced by different phytohormones. Among these phytohormone-responsive elements, the ABRE and MeJARE elements showed higher frequency, whereas AuxRE and GARE had a lower frequency in the promoters of *BnaOBMP* genes (Figure 4). Among the cis-acting regulatory elements, LTR (light-responsive element) exhibited the most abundance in the promoters of *BnaOBMP* genes. In addition, diverse cis-acting regulatory elements involved in abiotic and biotic stress-responsive elements, such as ARE (anoxic-responsive element), DSRE (defense- and stress-responsive element), DIRE (drought-responsive element), DRE (damage-responsive element), LTRE (low-temperature-responsive element) and WRE (wound-responsive element), were also identified in the promoter regions of *BnaOBMP* genes.

### 2.5. Extensive Sequence Polymorphisms of the BnaOBMP Genes

DNA sequence polymorphisms in the gene can provide insights into the evolutionary forces acting on populations and species adapting to different environments [40]. Based on the genomic resequencing dataset of worldwide rapeseed accessions [41], we performed a variation analysis on the *BnaOBMP* genes to assess their sequence polymorphisms. The polymorphism sites of exons and introns in the *BnaOBMP* genes ranged from 2 (*BnaOBMP.T15*) to 86 (*BnaOBMP.T26*) and 2 (*BnaOBMP.SH8*) to 951 (*BnaOBMP.S1*), respectively (Appendix A). Among the *BnaOBMP* genes, twelve genes contained 1 to 5 splicing variants, twenty-seven genes had one to three stop-gain variants, and four genes harbored one stop-loss variant, which indicates that these genes exhibited loss of function (Appendix A). Among the *BnaOBMP* genes, *BnaOBMP.U9* exhibited the highest coding sequence variation ratio of 0.167, while *BnaOBMP.T15* had the lowest variation ratio of 0.003 (Appendix A). The pi (π) values of the nucleotide diversity parameters extended from 0.000087 (*BnaOBMP.T16*) to 0.01528 (*BnaOBMP.T9*). The *BnaOBMP* genes of *BnaOBMP.T9* (π = 0.01528), *BnaOBMP.U9* (π = 0.00986), *BnaOBMP.T14* (π = 0.00871), *BnaOBMP.U7* (π = 0.00834), *BnaOBMP.C4* (π = 0.00772) and *BnaOBMP.SL3* (π = 0.00764) showed relatively higher nucleotide diversity (Figure 5a). In addition, the π values of *BnaOBMP* genes derived from the A and C subgenomes showed no significant difference (Student’s *t*-test, *p*-value = 0.055). To study the population selection pressures for *BnaOBMP* genes, we conducted neutrality test statistics using Tajima’s D-test method. Of all the *BnaOBMP* genes, approximately 27.3% (24/88) had positive Tajima’s D values, while the others were negative (Figure 5b). Notably, Tajima’s D values for four *BnaOBMP* genes, including *BnaOBMP.S13* (2.48971), *BnaOBMP.T14* (2.74572), *BnaOBMP.U7* (2.7529) and *BnaOBMP.C7* (4.01958), were over 2, while Tajima’s D values for eight *BnaOBMP* genes, including *BnaOBMP.SL4* (−2.14472), *BnaOBMP.SH8* (−2.04585), *BnaOBMP.C19* (−2.1825), *BnaOBMP.T15* (−2.10059), *BnaOBMP.T21* (−2.05404), *BnaOBMP.T22* (−2.00284), *BnaOBMP.T24* (−2.08246) and *BnaOBMP.T25* (−2.247), were less than −2, suggesting that these genes were under strong selection (positive or negative). The selection pressures on the twelve *BnaOBMPs* in rapeseed were also explored based on the *Ka*/*Ks* ratios. A *Ka*/*Ks* ratio greater than 1, equal to 1 and less than 1 indicated positive selection, neutral evolution and purifying selection at a low evolutionary rate, respectively. Among these *BnaOBMP* genes, except for *BnaOBMP.S13*, the *Ka*/*Ks* ratio values of the others were all less than 1 (Appendix A).

### 2.6. Diverse Expression Patterns of BnaOBMP Family Genes

To examine the expression patterns of the *BnaOBMP* genes during growth and development in rapeseed, we reanalyzed the publicly available RNA-Seq datasets (database accession: PRJNA394926 and PRJNA311067), which were obtained from eight tissues at different developmental stages including root (seedling), stem (seedling), leaf (mature), flower bud, sepal, stamen, pistil (new, blossoming and wilting) and seed (2, 4, 6 and 8 weeks after pollination (WAP)). Diverse expression patterns were observed for different subfamilies of *BnaOBMP* genes in rapeseed (Figure 6a). Of the caleosin genes, five members (i.e., *BnaOBMP.C1*, *BnaOBMP*.*C9*, *BnaOBMP*.*C10*, *BnaOBMP*.*C19* and *BnaOBMP*.*C20*) were specifically expressed in seed (6 and 8 WAP), four members (i.e., *BnaOBMP*.*C7*, *BnaOBMP*.*C8*, *BnaOBMP*.*C16* and *BnaOBMP*.*C17*) exhibited relatively high expression levels in root, five members (i.e., *BnaOBMP*.*C2*, *BnaOBMP*.*C3*, *BnaOBMP*.*C11*, *BnaOBMP*.*C12* and *BnaOBMP*.*C18*) were highly expressed in floral organs, while six members (i.e., *BnaOBMP*.*C4*, *BnaOBMP*.*C5*, *BnaOBMP*.*C6*, *BnaOBMP*.*C13*, *BnaOBMP*.*C14* and *BnaOBMP*.*C15*) displayed higher expression level in mature leaf of rapeseed. For the steroleosin genes, all members were preferentially expressed in seed, except for *BnaOBMP.S1* in pistil, *BnaOBMP.S8* in flower bud and *BnaOBMP.S15* in root. In addition, *BnaOBMP.S7* and *BnaOBMP.S9* were unexpressed in all tissues examined. For the oleosin genes, all the SH, SL and U oleosin genes were specifically expressed in seed, except for *BnaOBMP.U1* and *BnaOBMP.U5*, which were expressed in flower bud, while all the T oleosins, except for *BnaOBMP.T1* and *BnaOBMP.T8*, were highly expressed in flower bud (Figure 6a).

We further examined the expression difference of *BnaOBMP* genes between young (at 20 days after flowering (DAF)) and mature (at 40 DAF) seeds among different rapeseed accessions using the publicly available RNA-Seq dataset (database accession: CRA003544) (Figure 6b). Principal component analysis (PCA) analysis revealed that the 560 seed samples could be clustered into two distinct groups corresponding to the 20 DAF and the 40 DAF groups (Appendix A). The 40 DAF group showed more divergent distribution compared to the 20 DAF group for PC1, indicating that the expression of *BnaOBMP* genes was more diverse in mature seeds of rapeseed. Approximately 79.5% (70/88) of *BnaOBMP* genes, including 13 caleosins, 42 oleosins and 15 steroleosins, showed low expression levels (fragments per kilobase of exon per million fragments mapped (FPKM) <10) in the young seed samples. In contrast, approximately 43.1% (38/88) of *BnaOBMP* genes of 7 caleosins, 26 oleosins and 5 steroleosins exhibited low expression levels in the mature seeds (Figure 6b). Obviously, the SH and SL oleosin genes showed the highest expression levels in the seeds of the worldwide collected rapeseed accessions at 40 DAF compared to those at 20 DAF (Figure 6b). Moreover, there were 28 *BnaOBMPs*, including 5 caleosins, 18 oleosins and 5 steroleosins, that seemed to be unexpressed in all the seed samples of rapeseed (Figure 6b).

### 2.7. Quantitative Real-Time PCR (qRT-PCR) Analysis of the BnaOBMP Genes

To further validate the functional roles of *BnaOBMPs* under abiotic stresses and during seed development, six of the *BnaOBMP* genes from different subfamilies were selected for examining expression levels using qRT-PCR in the cultivated species of rapeseed (ZS11). These genes included two caleosins (i.e., *BnaOBMP.C19* and *BnaOBMP.C7*), three oleosins (i.e., *BnaOBMP.SH8*, *BnaOBMP.SL4* and *BnaOBMP.U9*) and one steroleosin (i.e., *BnaOBMP.S13*). After a 3 h heat treatment, the expression level of *BnaOBMP.C7* was significantly downregulated by approximately five-fold compared with the control, while *BnaOBMP.SH8* was significantly upregulated in the leaves of rapeseed (Figure 7). After a 3 day drought treatment, the expression levels of *BnaOBMP.C7*, *BnaOBMP.U9* and *BnaOBMP.SH8* were reduced (Figure 7). In contrast, *BnaOBMP.C19*, *BnaOBMP.SL4* and *BnaOBMP.S13* showed no detectable expression in rapeseed leaves. Compared to seeds of 15 DAF (early seed development), the expression levels of *BnaOBMP.C19*, *BnaOBMP.SH8*, *BnaOBMP.S13* and *BnaOBMP.U9* increased in seeds of 25 DAF (seed filling stage) and reached a peak in seeds of 35 DAF (seed filling stage) and then decreased in seeds of 50 DAF (seed maturation stage). Both *BnaOBMP.C7* and *BnaOBMP.SL4* exhibited significantly induced expression during the development of seed (Figure 7). qRT-PCR analysis validated the gene expression patterns identified by RNA-seq.

### 2.8. Co-Expression and Gene Regulatory Network of BnaOBMPs during Seed Development

In order to comprehensively analyze the gene interactions and regulatory relationships of *OBMP* genes during seed development, the previous seed RNA-seq data were employed to build a co-expression and gene regulatory network. We calculated the Pearson correlation coefficients (PCCs) of the expression levels between *BnaOBMP* gene pairs (Appendix A). As the gene expression correlation matrix of the 88 *BnaOBMP* genes shows, the caleosins of *BnaOBMP.C3*, *BnaOBMP.C4*, *BnaOBMP.C5*, *BnaOBMP.C12*, *BnaOBMP.C15*, *BnaOBMP.C16* and *BnaOBMP.C18* and the T oleosins of *BnaOBMP.T11*, *BnaOBMP.T12*, *BnaOBMP.T14*, *BnaOBMP.T24*, *BnaOBMP.T25*, *BnaOBMP.T26* and *BnaOBMP.T27* had negative correlations with most of the other *BnaOBMP* genes, while the rest of the genes exhibited positive correlations with each other (Appendix A). Subsequently, all gene pairs with significant PCCs (*p*-value ≤ 0.01 and |PCC| > 0.6) were extracted and used to construct co-expression networks. The co-expression networks of *BnaOBMPs* were constituted with 602 edges and 56 nodes including 38 oleosins, 8 caleosins and 10 steroleosins (Figure 8a). There were twenty-eight *BnaOBMP* gene pairs showing negative correlations (*p*-value ≤ 0.01 and PCC < −0.6), which were between *BnaOBMP.C15* and other *BnaOBMPs* including *BnaOBMP.C1*, *BnaOBMP.C10*, *BnaOBMP.C19*, *BnaOBMP.C20*, *BnaOBMP.S2*, *BnaOBMP.S6*, *BnaOBMP.S10*, *BnaOBMP.S14*, *BnaOBMP.SH1*, *BnaOBMP.SH2*, *BnaOBMP.SH3*, *BnaOBMP.SH4*, *BnaOBMP.SH5*, *BnaOBMP.SH6*, *BnaOBMP.SH7*, *BnaOBMP.SH8*, *BnaOBMP.SH9*, *BnaOBMP.SL1*, *BnaOBMP.SL2*, *BnaOBMP.SL3*, *BnaOBMP.SL4*, *BnaOBMP.SL5*, *BnaOBMP.SL6*, *BnaOBMP.SH7*, *BnaOBMP.U2*, *BnaOBMP.U4*, *BnaOBMP.U6*, *BnaOBMP.U8* and *BnaOBMP.U9*, while a total of 574 *BnaOBMP* gene pairs displayed positive correlations. Particularly, over a half of these gene pairs (316/574) exhibited strong positive correlations (*p*-value ≤ 0.01 and PCC > 0.8).

To explore transcription factors involved in the regulation of *OBMP* genes during seed development, we further constructed a seed-specific gene regulatory network (GRN) using GENIE3 algorithm in *B. napus*. By filtering for the genes expressed > 1 FPKM in at least one of the seed samples, we captured 61,907 genes, including 4643 transcription factors (TFs) and 64 *BnaOBMP* genes, for GRN construction. The top 5 TFs of each *BnaOBMP* connected were extracted from the GRN and used to construct a subnetwork of *BnaOBMP*-GRN (Figure 8b). The classification of these TFs in *BnaOBMP*-GRN revealed that the expression of *BnaOBMP* genes were potentially regulated by various types of TFs such as ZIP, C2C2-GATA, HB-HD-ZIP, HB-WOX, C2H2, NF-YB, C3H, bHLH, AP2/ERF-ERF, NAC, GARP-G2-like, MYB and C2C2-dof (Figure 8b). Notably, a total of 159 hub TFs were identified in *BnaOBMP*-GRN (Appendix A). For examples, the bZIP gene of *BnaA04G0262900ZS* encoded ABSCISIC ACID-INSENSITIVE 5-like protein 3, had the greatest hub value, the candidate targets of which were *BnaOBMP.S5*, *BnaOBMP.SH3*, *BnaOBMP.SL7*, *BnaOBMP.SL3*, *BnaOBMP.SH9* and *BnaOBMP.U9*. In addition, the C2C2-GATA gene of *BnaA08G0133800ZS* encoded GATA transcription factor 8-like protein, which might regulate the expression of *BnaOBMP.SH3*, *BnaOBMP.SH6*, *BnaOBMP.SL3*, *BnaOBMP.SL8*, *BnaOBMP.C17* and *BnaOBMP.S6*.

## 3. Discussion

OBs share common features in all kingdoms of life [42], consisting of a densely packed hydrophobic core of neutral lipids enclosed by a phospholipid monolayer decorated by three main classes of oil-body-membrane proteins (OBMP) including oleosins, caleosins and steroleosins [13,42]. Oleosins are the most abundant protein constituents and are sufficient to cover the whole surface of a seed OB [43]. Oleosins play important roles in the formation and stabilization of OBs during seed and pollen grain development as well as in OB turnover [26]. Caleosin is a common OB surface protein found in a wide range of plant species [44]. Caleosin has a Ca^2+^-binding motif, which has the ability to bind calcium. Recent studies have suggested that caleosins also possess peroxygenase activities that convert hydroperoxides of α-linolenic acid to various oxylipins as phytoalexins [27,45]. Steroleosins belong to hydroxysteroid dehydrogenases (HSDs), consisting of a sterol-binding site and an NADPH-binding site involved in some biological functions related to membrane remodeling and lipid signaling [32,46,47]. Thus, OBMP is not only the key structural molecule for the formation and stabilization of OBs but may also exert a myriad of cellular functions related to carbon, energy and lipid metabolisms; stress responses; hormone signaling pathways; be involved in various aspects of plant growth and development.

As the second-largest source of vegetable oil, rapeseed is an important worldwide oilseed crop [2]. Their seeds contain lipids as major storage reserves, which is up to 50% of the dry weight, and the main component of lipids is triacylglycerol stored in oil bodies [48]. To investigate the function of oil-body-membrane proteins from important oil crops, we performed identification and characterization of the *OBMP* gene family in the polyploid crop *B. napus* in the present study. A total of 88 *OBMP* genes were found in all of the nineteen chromosomes of rapeseed, which were classed into 53 oleosins (27 T, 8 SL, 9 SH and 9 U), 20 caleosins and 15 steroleosins based on their functional domains and phylogenetic relationships (Table 1). Compared to the number of *OBMP* genes identified in the species from green algae to higher plants, *B. napus* contained the most abundant *OBMP* genes, suggesting the massive expansion of this gene family in rapeseed (Appendix A). The A and C subgenomes of *B. napus* contained 43 and 45 *BnaOBMP* genes, respectively, which is comparable to the 44 and 47 genes identified in *B. rapa* and *B. oleracea*, respectively (Table 1 and Appendix A). Moreover, the *BnaOBMP* genes from the A and C subgenomes showed conserved synteny and gene order (Figure 3). These results indicate that *B. napus* retained the vast majority of *OBMP* genes from its two ancestors during the allopolyploidization.

Gene and genome duplications gave rise to the number of genes, resulting in functional redundancy and differentiation of genes, enabling genome-wide adaptation to various environments during evolution [37]. The previous study revealed that the crucifer (*Brassicaceae*) lineage experienced two whole-genome duplications (WGDs) and one whole-genome triplication event (WGT), shared by most dicots [49]. Moreover, the *Brassica* species experienced an extra WGT event approximately 15.9 million years ago compared with *A. thaliana* [50]. *B. napus* is a relatively new species of the *Brassica* genus, with a short history of post-Neolithic speciation (~7500 years) and domestication (~700 years). Consistent with these WGD and WGT events, the *OBMP* genes experienced gene duplication events leading to an expanded *OBMP* gene family in rapeseed. The WGD or segmentally duplicated genes accounted for the majority of *BnaOBMP* gene family. Particularly, all of the caleosins and steroleosins resulted from WGD or segmental duplication in rapeseed. The tandemly duplicated genes were also detected in the *BnaOBMP* gene family, and all fifteen tandemly duplicated genes belonged to T oleosins, which might be the result of the aggregated distribution of T oleosins in the chromosomes of the rapeseed genome. Furthermore, the *BnaOBMP* genes showed different levels of polymorphism. Most of the *BnaOBMP* genes had *Ka*/*Ks* ratios less than 1 and Tajima’s D values less than 0, suggesting that the *BnaOBMP* gene family experienced strong purifying (stabilizing) selection rather than positive selection during the evolution. In addition, the *Ka*/*Ks* ratios were substantially highest among the T oleosins than other *BnaOBMP* genes, implying that T oleosins evolved faster than the other *BnaOBMP* genes in rapeseed. Altogether, these results revealed that the *BnaOBMP* gene family expanded primarily by gene duplications with WGD/segmental duplication being the major driving force in *B. napus*.

Oil bodies, as essential lipid storage organelles in the seeds of plants, play important roles in seed germination and the postgerminative growth of seedlings, as well as many other cellular processes such as stress responses, lipid metabolism, organ development, and hormone signaling. These biological functions of seed OBs depend on OBMP proteins, which are embedded in the OB phospholipid monolayer. Our results revealed that various cis-acting regulatory elements exist in the promoters of *BnaOBMP* genes. The cis-acting regulatory elements exert various functions associated with plant growth and development, phytohormone responsive, and abiotic and biotic stress responsive such as ABRE (abscisic acid-responsive element), AuxRE (auxin-responsive element), ERE (ethylene-responsive element), ARE (anoxic-responsive element), DIRE (drought-responsive element) and LTRE (low-temperature-responsive element). This suggests that *BnaOBMP* genes could be induced by different phytohormone and stress signals so as to adjust OBs to different environmental conditions.

Unraveling the expression pattern of different OB proteins throughout seed development is crucial for improving our understanding of OB formation. A previous study revealed that accumulation of oleosins S1-S5 and caleosin CLO1 began at approximately 12 days after pollination (DAP), while steroleosin SLO1 accumulated later at approximately 25 DAP, and then they all increased rapidly and reached a peak at 55–60 DAP in *A. thaliana*, as analyzed by immunoblot [51]. We found that the expression of most *BnaOBMP* genes were upregulated along the development of the seed and showed the highest expression levels at the late stages, which is consistent with the previous study. Although OBs occur minimally in nonstorage vegetative organs, we also observed that some *BnaOBMP* genes exhibited high expression in flower bud, pistils, leaves, stems and roots, indicating that OBs are present in these tissues. For example, T oleosins, except for *BnaOBMP.T1* and *BnaOBMP.T8*, were preferentially highly expressed in flower bud. Some caleosins showed higher expression levels in flower bud, stamen, pistil, leaf and root. For steroleosins, *BnaOBMP.S1* showed the highest expression level in pistil, while *BnaOBMP.S15* was highly expressed in root. These results suggested that OBs could be not only present in seed as storage warehouses but also exist in nonstorage vegetative organs as detoxification refuges.

Genes involved in the same process usually have similar expression patterns, and they typically grouped into the same module in co-expression analysis. Here, based on the co-expression networks of *BnaOBMPs*, the *BnaOBMP* family genes showed a similar tendency and gathered closely together in one cluster during seed development in *B. napus*. Strong positive correlations were observed between the members of the *BnaOBMP* gene family. Meanwhile, only a few significant negative correlations appeared between *BnaOBMP.C15* and the others. The results suggested that the *BnaOBMP* genes might share functional redundancy in *B. napus*. To discern gene transcriptional regulatory mechanisms of *BnaOBMP* genes, we constructed a regulatory network incorporating TF information by GENIE3. Previous studies demonstrated that a GENIE3 network could provide biologically relevant transcription factor-target relationships in wheat [52,53]. Our *BnaOBMP*-GRN network revealed that the *BnaOBMP* genes could be regulated by various transcript factors. After prioritization of the candidate regulatory genes, the top hub transcription factor was the bZIP gene of *BnaA04G0262900ZS* encoded ABSCISIC ACID-INSENSITIVE 5 (ABI5)-like protein 3 (EEL). The *Arabidopsis* EEL (known as AtbZIP12) is transcription factor homologous to ABI5, which is a key player in light-, abscisic acid-, and gibberellin-signaling pathways to precisely control seed maturation and germination [54,55,56]. In addition, some other hub TFs, such as GATA3, HAT2, SMZ, DOF5.6 and APL, might also play important roles in the regulation of *BnaOBMPs* during OB formation in rapeseed, which needs further investigation.

In conclusion, our results reveal that *B. napus* had an expansion of the *OBMP* gene family due to the fact of WGD and tandem duplications. These *BnaOBMP* genes contain extensive sequence polymorphisms, and some members may have experienced strong selection. Various cis-acting regulatory elements involved in plant growth, phytohormone and abiotic and biotic stress responses were found in their promoter regions. In addition, both transcriptomic and qRT-PCR analyses corroborated that *BnaOBMPs* exhibited spatiotemporal expression patterns and are preferentially expressed in seeds. The genetic variations (i.e., SNPs or InDels) of *BnaOBMP* genes can be used as molecular markers to select rapeseed cultivars with high seed oil content (SOC). Moreover, further manipulating the expression patterns of some candidate *BnaOBMPs* during seed development using genetic engineering techniques, such as transgenic technology and CRISPR/Cas9 tools, would contribute to the increase in the SOC in rapeseed.

## 4. Materials and Methods

### 4.1. Identification and Property Analysis of BnaOBMP Genes

The *OBMP* genes in *A. thaliana* were retrieved from the TAIR (http://www.arabidopsis.org/ (accessed on 7 January 2022) database. The protein sequences, coding sequences (CDS), genome sequences and annotation of *Brassica napus* var. ZS11 were obtained from BnPIR database (http://cbi.hzau.edu.cn/bnapus/ (accessed on 8 January 2022). The protein sequences of *B. rapa* var. Z1 and *B. oleracea* var. HDEM were downloaded from the GENOSCOPE database (http://www.genoscope.cns.fr/ (accessed on 8 January 2022). The protein sequences of the other 51 species were obtained from the NCBI database (https://www.ncbi.nlm.nih.gov/ (accessed on 12 January 2022). The peptides of the OBMP proteins of *A. thaliana* were used as queries in a BLASTP (version 2.2.26) search against all the annotated protein sequences of *B. napus* with an E-value threshold of 1 × 10^−5^. Then, the sequences of the predicted BnaOBMP proteins were searched against all the annotated proteins of *A. thaliana*. The putative BnaOBMPs with the best hits on *A. thaliana* OBMP proteins remained. For further confirmation of BnaOBMP proteins, the oleosin domain (PF01277), the caleosin domain (PF05042) or the steroleosin domains (PF00106 or PF13561) from the Pfam database (http://pfam.xfam.org/ (accessed on 17 February 2022) were applied as queries to search against the putative BnaOBMP protein sequences using HMMER (version 3.2.1) with anE-value setting of 1 × 10^−5^. The OBMP genes of other species were also identified in this way. The properties of each BnaOBMP, including molecular weight (MW), isoelectric point (pI), instability index, aliphatic index and grand average of hydropathy (GRAVY), were calculated using the ProtParam tool (https://web.expasy.org/protparam/ (accessed on 9 March 2022). The subcellular localization of each BnaOBMP was predicted by LOCALIZER (https://localizer.csiro.au/ (accessed on 28 March 2022).

### 4.2. Gene Structure and Chromosomal Localization Analysis

Based on the genome annotation of *B. napus* var. ZS11, available from the BnPIR database (http://cbi.hzau.edu.cn/bnapus/ (accessed on 8 January 2022)), a graphical representation of the exon–intron structure of each BnaOBMP gene was drawn using Gene Structure Display Server 2.0 (http://gsds.cbi.pku.edu.cn/ (accessed on 12 March 2022)). The schematic map of the *BnaOBMP* genes on chromosomes was plotted using R software with RIdeogram package (version 0.2.2) according to their physical chromosomal locations on *B. napus*.

### 4.3. Cis-Acting Regulatory Elements and Motif Analysis

The 2 Kb sequence upstream from the transcription start site of each *BnaOBMP* gene was defined as the promoter region and was extracted from the *B. napus* genome sequence. The cis-acting regulatory elements within the promoters were analyzed by PlantCARE (https://bioinformatics.psb.ugent.be/webtools/plantcare/html/ (accessed on 23 March 2022)).

### 4.4. Multiple Sequence Alignments and Conserved Motif Analysis

MUSCLE (version 3.8) was used to align the OBMP protein sequences of *B. napus*, *B. rapa*, *B. oleracea* and *A. thaliana* with default parameters. The MEME suite (https://meme-suite.org/meme/ (accessed on 14 April 2022)) was employed to identify the motifs with conserved amino acids in the protein sequences of *BnaOBMP**s* with default parameters.

### 4.5. Phylogenetic Analysis

Based on the multiple sequence alignments of OBMP proteins from *B. napus*, *B. rapa*, *B. oleracea* and *A. thaliana*, IQ-TREE (version 2.0.3) was employed to reconstruct a maximum likelihood (ML) gene tree with 1000 replicates. The VT + F + R4 model was the best-fit evolutionary model selected by ModelFinder implemented in IQ-TREE. The obtained ML gene trees were visualized using iTOL (https://itol.embl.de/ (accessed on 24 March 2022)).

### 4.6. Synteny and Duplicate Gene Analysis

The syntenic blocks and gene duplications were identified within the *B. napus*, *B. rapa*, *B. oleracea* and *A. thaliana* genomes using MCScanX (version 3.8.31) with the default parameters. The circos graph with the genomic collinearity of *BnaOBMP* genes was plotted using the Rcircos package in R software (version 3.6.1).

### 4.7. Sequence Variations and Polymorphism Analysis

The publicly available genomic resequencing dataset of different *B. napus* accessions from around the world (database accession: SRP155312) was collected from the National Center of Biotechnology Information (NCBI) database [41]. The alignment and variant calling were performed using Sentieon DNASeq Variant Calling Workflow [57]. Variant annotation was achieved by ANNOVAR (version 20220320) based on the annotation of the *B. napus* var. ZS11 genome. The sequence diversity metrics, including pairwise nucleotide variation as a measure of variability (π) and selection statistics (Tajima’s D) of *BnaOBMP* genes, were calculated by vcftools (version 0.1.13) based on SNP distribution.

### 4.8. Expression Analysis of BnaOBMP Genes

The publicly available RNA-seq datasets (database accession: PRJNA394926 and PRJNA311067) of eight tissues (i.e., root, stem, leaf, flower bud, sepal, stamen, pistil and seed) collected from different developmental stages [58,59] and an RNA-Seq dataset (database accession: CRA003544) of 20 DAF and 40 DAF seeds of 280 *B. napus* germplasm accessions [60] were downloaded from the NCBI and NGDC databases, and they were used for gene expression profiling. Mapping of these RNA-Seq reads against the *B. napus* var. ZS11 reference genome using HISAT2 (version 2.1.0) with the default settings. The reads count per gene was calculated with HTSeq (version 0.9.172) and was further used to calculate the FPKM values for the quantification of gene expression. The heatmaps were visualized using the pheatmap package (version 1.0.12) in R software (version 3.6.1). The principal component analysis was performed using the function prcomp() in the R software.

### 4.9. Plant Materials and Treatments

The seeds of the rapeseed cultivar ZS11 were germinated on a filter paper saturated with distilled water in darkness at 22 °C for 3 days. The seedling plants were transplanted to soil culture pots in a greenhouse to grow for six weeks under well-controlled conditions as follows: a temperature of 25 °C, light intensity of 150 μmol m^−2^s^−1^ provided by a high-pressure sodium lamp, and a humidity of 50–60%. Then, the rapeseed plants before the bolting stage were chosen for analysis. The top third of fully expanded leaves from rapeseed plants were sampled as a control before being stressed for the following experiments. The drought and heat stress treatments were conducted in plant growth chambers with well-controlled temperature and humidity. For the drought treatment, water was withdrawn for 7 days, and then the plants were rewatered to recover from the stress. The growth chamber was programmed as follows: 40% humidity in 16 h light at a temperature of 25 °C; 45% humidity in 8 h dark at a lower temperature of 22 °C. Leaf samples were collected 3 days after drought treatment. For heat treatment, the growth chamber was set at 60% humidity in 16 h light at a high temperature of 40 °C, followed with 55% humidity in 8 h dark at a temperature of 35 °C. Heat-treated leaf samples were collected at a time point of 3 h during the stress treatment. Rapeseed seeds at 15, 25, 35 and 50 days after flowering of the ZS11 plants cultivated in the experimental field were sampled for analysis. All samples were immediately frozen using liquid nitrogen after being detached, and they were stored at −80 °C for further assay.

### 4.10. RNA Isolation and qRT-PCR Analysis

Total RNA was extracted from each sample using an RNA extraction kit (Takara, Dalian, China) following the manufacturer’s procedure. Two micrograms of total RNA were used to synthesize the first-strand cDNA using the Prime Script RT reagent Kit (Takara, Dalian, China) according to the manufacturer’s protocol. Quantitative real-time PCR was performed using 2 μL of cDNA in a 20 μL reaction volume with SYBR Premix Ex Taq (Takara) on a 7500-Fast real-time PCR System (Applied Biosystems). Gene-specific primers were designed and are listed in the Appendix A). The thermal cycler was set as follows: an initial incubation at 50 °C for 2 min and 95 °C for 5 min, followed by 40 cycles at 95 °C for 30 s, 55 °C for 30 s and 72 °C for 30 s. All qRT-PCR reactions were assayed in triplicates. The relative quantification of transcription level was determined by the methods described previously [61].

### 4.11. Pearson Correlation and Gene Regulatory Network Analysis

Based on the previous seed RNA-seq (database accession: CRA003544) analysis results, the PCC values and significant of correlations among *BnaOBMP* genes were calculated using the function rcorr() of R software (version 3.6.1). The genes with FPKM >1 in at least one of the 560 seed samples of *B. napus* were selected and used to construct the gene regulatory network (GRN) using GENIE3 (version 1.19.0). The TFs of *B. napus* identified by iTAK (version 1.7) were used as candidate regulators for the GRN construction. Gephi (version 0.9.2) was used to calculate network metrics and create visualization graphs.

## Figures and Tables

**Figure 1 plants-11-02241-f001:**
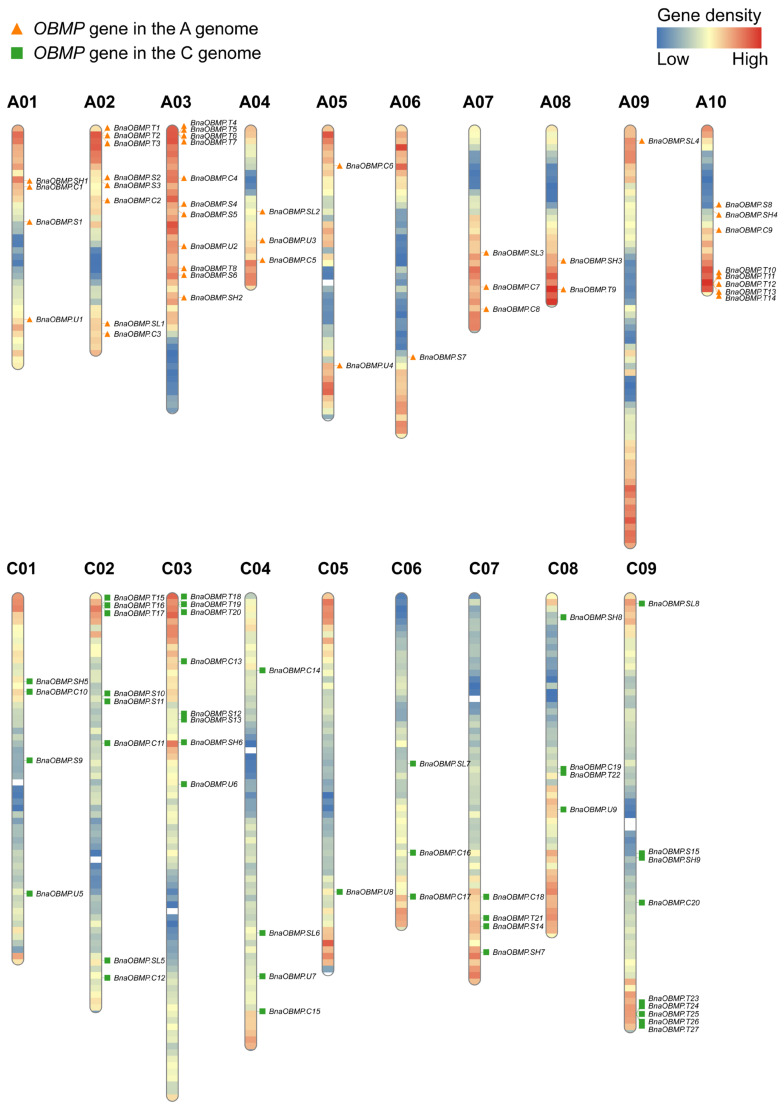
Genomic distributions of *BnaOBMP* genes on *B. napus* chromosomes. The *BnaOBMPs* were plotted based on the location of the genes and the length of the chromosomes. The orange triangles and green squares represent *BnaOBMP* genes in the A and C subgenomes, respectively. The color on each chromosome denotes the gene density from low (blue) to high (red) by the frequency per 1 mega base (Mb).

**Figure 2 plants-11-02241-f002:**
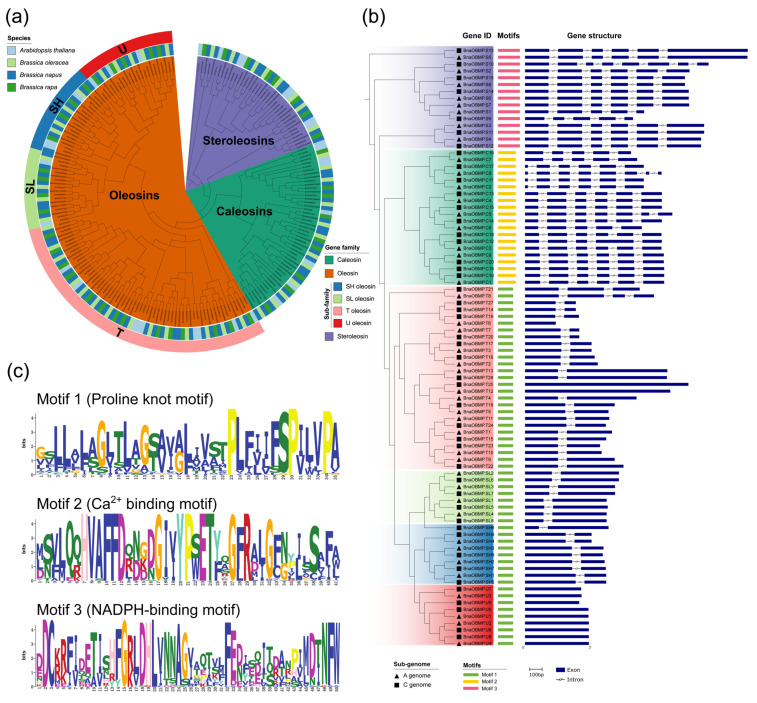
Phylogenetic relationships, motifs and gene structures of *BnaOBMPs*. (**a**) Phylogenetic relationships of *BnaOBMPs*, *BraOBMPs*, *BolOBMPs* and *AtOBMPs*. The unrooted tree was generated using IQ-TREE software by the ML method. The outermost track represents the subfamily of oleosins including U (red), SH (blue), ST (light green) and T (pink) oleosins. The middle track marks the species by the corresponding colors shown in the color legend at the top left. The inner track represent the ML tree and the phylogenetic classes of *OBMP* genes including oleosins (orange), caleosins (green) and steroleosins (light blue). The color legend for the *OBMP* gene family is shown at the bottom right. (**b**) A schematic diagram of the exon–intron organization of *OBMP* genes in *B. napus*. The phylogenetic tree of *BnaOBMPs* is placed at the left, and the background colors represent phylogenetic classes. The black triangles and squares represent *BnaOBMP* genes in the A and C sub genomes of *B. napus*, respectively. The blue boxes and shrunken lines indicate exons and introns, respectively. The length scale bar is 100 bp. The diverse conserved motifs of *BnaOBMPs* are marked by different colored boxes including Motif 1 (green), Motif 2 (yellow) and Motif 3 (pink). (**c**) The conserved domain sequence model of Motifs 1, 2 and 3. Motif 1 is a proline knot motif including four invariable residues of the proline knot sequence (-PX_5_SPX_3_P-). Motif 2 is a Ca^2+^-binding motif. Motif 3 is an NADPH-binding motif.

**Figure 3 plants-11-02241-f003:**
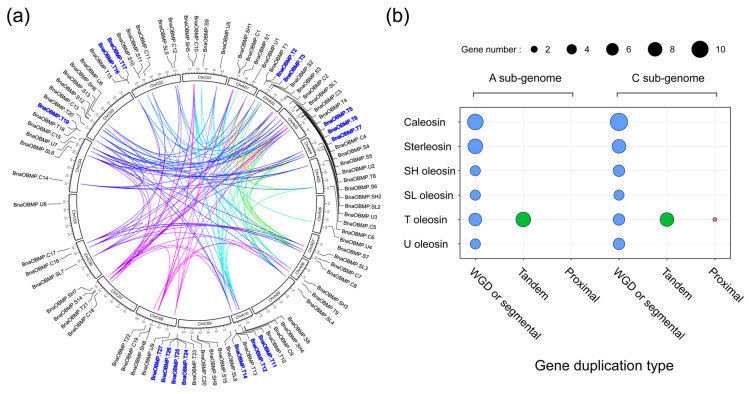
Collinear correlations and gene duplication type of the *BnaOBMP* family genes. (**a**) Collinear correlations of *OBMP* genes in *B. napus*. The circle boxes represent chromosomes with names. The scale on the circle is in mega bases. *BnaOBMP* gene IDs on the chromosomes indicate their physical positions. The inner lines indicate the collinear correlations between *BnaOBMP* gene pairs. Gene IDs marked by a blue font signify tandem duplication clusters. (**b**) The gene numbers for each family of *BnaOBMPs* with different gene duplication types. The *x*- and *y*-axes represent the gene duplication type and *OBMP* gene subfamilies, respectively. The size of the circles indicates the gene number. The A and C subgenomes are shown in the left and right panels, respectively.

**Figure 4 plants-11-02241-f004:**
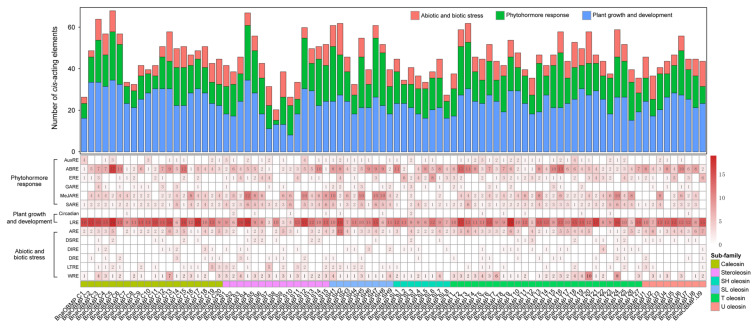
Cis-acting regulatory elements in the promoter region of *BnaOBMP* Genes. The upper panel of the bar plot represents the total number of cis-acting regulatory elements in each gene promoter region. The bars with different colors represent different types of cis-acting regulatory elements involving in abiotic and biotic stress response (red), phytohormone response (green) and plant growth and development (blue). The lower panel of the heatmap indicates the numbers of various cis-acting regulatory elements in these *BnaOBMP* genes. The color intensity of the cell denotes the number from low (white) to high (red). The numbers are displayed in the cells. The different *OBMP* gene subfamilies are marked by corresponding colors that are shown in the color legend at the bottom right.

**Figure 5 plants-11-02241-f005:**
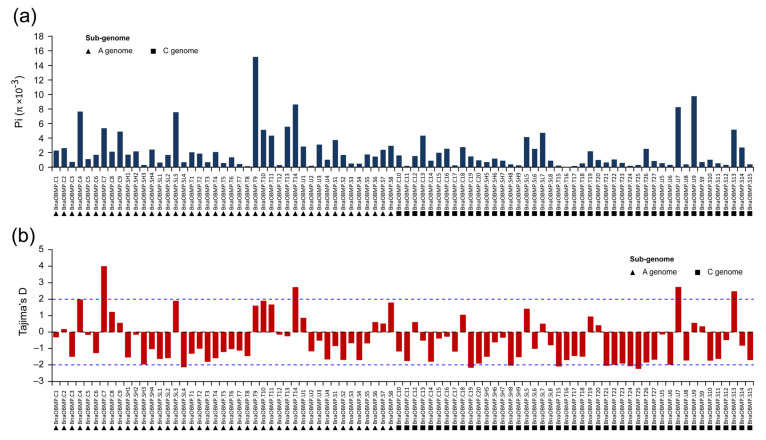
Observed pi (π) and Tajima’s D values for *BnaOBMP* genes. (**a**) Pi (π) values for each *BnaOBMP* gene. The black triangles and squares represent *BnaOBMP* genes in the A and C subgenomes of *B. napus*, respectively. (**b**) Tajima’s D values for the *BnaOBMP* genes. The *BnaOBMP* genes derived from the A subgenome and C subgenome are marked by black triangles and squares, respectively. The blue, dashed lines represent the cutoff of 2 and −2. The black, dashed line represents the baseline of 0.

**Figure 6 plants-11-02241-f006:**
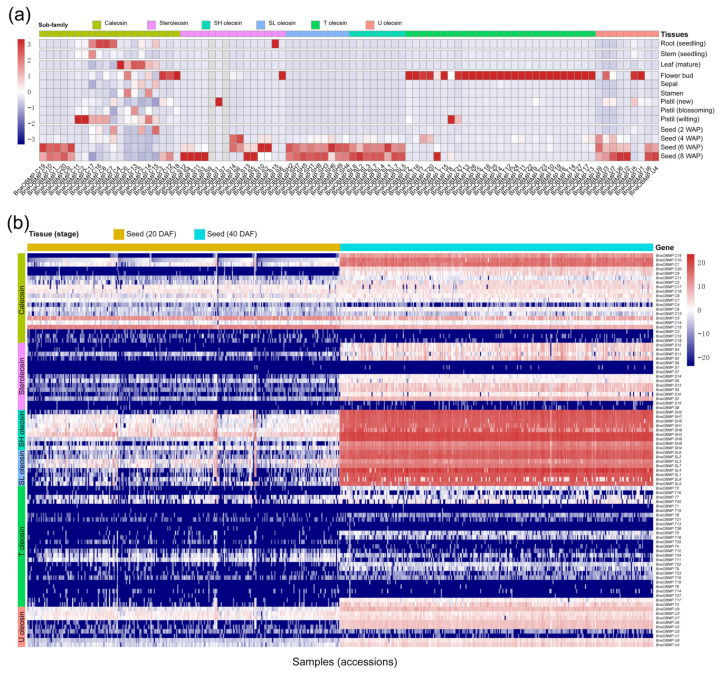
Expression analysis of the *BnaOBMP* genes in *B. napus*. (**a**) Expression patterns of the *BnaOBMP* genes in various tissues at different developmental stages. The color of each cell indicate the expression level of each *BnaOBMP* gene from low (blue) to high (red), expressed as log_10_FPKM. The *OBMP* gene subfamilies are marked by different colors that are shown in the color legend at the top. The tissues are listed at the right, with stages noted in the parentheses. (**b**) Expression patterns of the *BnaOBMP* genes in 20 DAF and 40 DAF seeds of 280 *B. napus* accessions. The 20 DAF (brown) and 40 DAF (cyan) seed samples of 280 *B. napus* accessions are displayed. Each *OBMP* gene subfamily is marked by the corresponding color on the left.

**Figure 7 plants-11-02241-f007:**
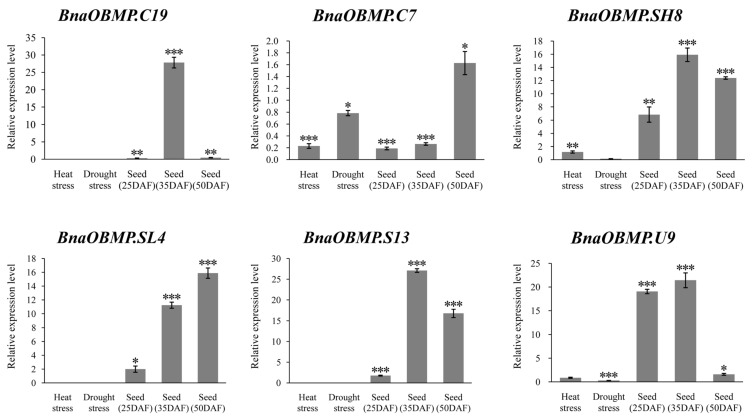
qRT-PCR verification that the expression of some representative *BnaOBMP* genes responded to heat and drought stresses (leaves) as well as during seed development. Statistically significant differences (Student’s *t*-test) are indicated as followed: * *p* < 0.05, ** *p* < 0.01, *** *p* < 0.001.

**Figure 8 plants-11-02241-f008:**
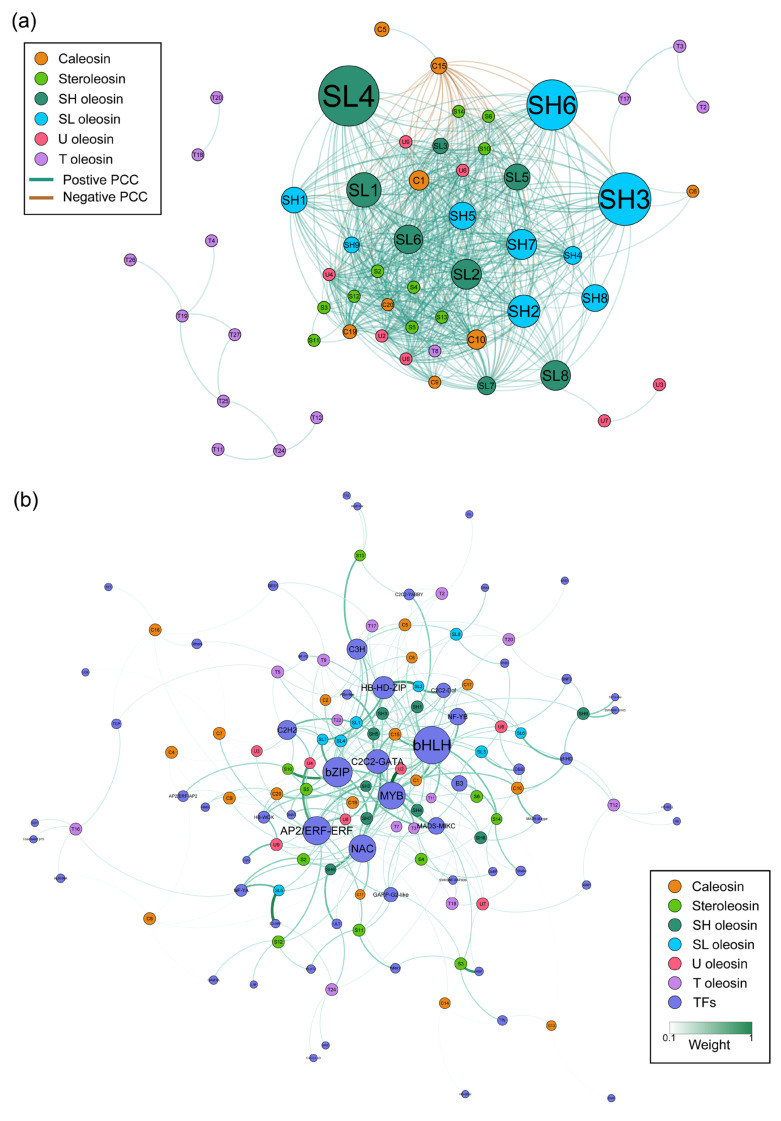
Co-expression and gene regulatory networks of *BnaOBMP* genes during seed development. (**a**) Co-expression network of *BnaOBMP* genes. The OBMP gene families are marked by different colors as shown in the color legend at the top left. The sizes of circles indicate the average expression levels of the *BnaOBMP* genes. The lines represent the positive (green) and negative (brown) correlations between *BnaOBMP* genes. (**b**) Gene regulatory networks of the *BnaOBMP* genes. The *OBMP* gene families and TFs are marked by corresponding colors shown in the color legend at the bottom right. The size of the circles indicates the degree of each node. The color and thickness of the lines represent the weight between TFs and *BnaOBMPs* from weak (white) to strong (green). The gene names are labeled on the nodes.

**Table 1 plants-11-02241-t001:** Summary information on *OBMP* family genes in *B. napus*.

Gene Name	Locus Name	Gene Location ^1^	mRNA (bp)	Protein (aa)	MW (kDa)	pI	Instability Index	Aliphatic Index	GRAVY	Homolog of *Arabidopsis*	Domain (Start–End aa)	Subcellular Localization ^2^
*BnaOBMP.C1*	*BnaA01G0161500ZS*	A01:9597423-9598812:−	738	245	28.12	5.81	51.68	80.53	−0.249	*AT4G26740*	PF05042 (64––230)	-
*BnaOBMP.C2*	*BnaA02G0189800ZS*	A02:11721396-11722848:+	579	192	21.50	9.38	36.29	71.72	−0.491	*AT1G70670*	PF05042 (16––182)	-
*BnaOBMP.C3*	*BnaA02G0368700ZS*	A02:32467336-32469224:−	717	238	27.11	7.77	40.61	64.83	−0.387	*AT5G29560*	PF05042 (60–227)	-
*BnaOBMP.C4*	*BnaA03G0160900ZS*	A03:8240819-8242440:−	720	239	26.86	4.93	43.03	77.53	−0.374	*AT2G33380*	PF05042 (59–225)	-
*BnaOBMP.C5*	*BnaA04G0214400ZS*	A04:20989630-20991888:−	732	243	27.32	5.2	45.42	74.28	−0.368	*AT2G33380*	PF05042 (58–229)	-
*BnaOBMP.C6*	*BnaA05G0108500ZS*	A05:6320856-6322131:+	633	210	23.59	5.98	52.05	73.43	−0.46	*AT2G33380*	PF05042 (83–196)	-
*BnaOBMP.C7*	*BnaA07G0266500ZS*	A07:25170417-25171498:−	597	198	22.41	9.28	39.75	71.97	−0.462	*AT1G70670*	PF05042 (22–188)	-
*BnaOBMP.C8*	*BnaA07G0320400ZS*	A07:28607934-28611160:+	579	192	21.43	7.1	40	75.21	−0.446	*AT1G70670*	PF05042 (16–175)	-
*BnaOBMP.C9*	*BnaA10G0113200ZS*	A10:16318187-16319594:+	735	244	27.96	5.74	48.36	75.61	−0.271	*AT5G55240*	PF05042 (63–228)	-
*BnaOBMP.C10*	*BnaC01G0206600ZS*	C01:15403062-15404481:−	738	245	28.13	5.81	50.55	80.53	−0.245	*AT4G26740*	PF05042 (64–230)	-
*BnaOBMP.C11*	*BnaC02G0251600ZS*	C02:23385393-23386865:+	579	192	21.51	9.44	36.64	71.72	−0.493	*AT1G70670*	PF05042 (16–182)	-
*BnaOBMP.C12*	*BnaC02G0494300ZS*	C02:59864850-59869109:−	789	262	30.05	8.4	41.25	69.27	−0.271	*AT5G29560*	PF05042 (60–151)	-
*BnaOBMP.C13*	*BnaC03G0187300ZS*	C03:10665491-10667041:−	720	239	26.82	5.18	47.02	77.95	−0.35	*AT2G33380*	PF05042 (58–225)	-
*BnaOBMP.C14*	*BnaC04G0135700ZS*	C04:12075783-12077313:+	720	239	26.94	5.3	48.29	73.89	−0.4	*AT2G33380*	PF05042 (58–225)	-
*BnaOBMP.C15*	*BnaC04G0525300ZS*	C04:65070023-65071948:−	720	239	26.92	5.11	46.4	75.52	−0.357	*AT2G33380*	PF05042 (58–225)	-
*BnaOBMP.C16*	*BnaC06G0299400ZS*	C06:40411272-40412854:−	549	182	20.44	9.34	38.27	71.32	−0.486	*AT1G70670*	PF05042 (22–150)	-
*BnaOBMP.C17*	*BnaC06G0373300ZS*	C06:47245132-47246570:+	579	192	21.63	7.13	40.97	71.67	−0.557	*AT1G70670*	PF05042 (16–182)	-
*BnaOBMP.C18*	*BnaC07G0337800ZS*	C07:47280426-47281567:−	717	238	27.12	7.72	40.64	63.24	−0.384	*AT5G29560*	PF05042 (60–227)	-
*BnaOBMP.C19*	*BnaC08G0170100ZS*	C08:28001595-28004342:−	738	245	28.06	5.65	50.17	84.49	−0.228	*AT4G26740*	PF05042 (64–230)	-
*BnaOBMP.C20*	*BnaC09G0377100ZS*	C09:48175369-48176860:−	735	244	28.21	5.9	45.12	76.02	−0.273	*AT5G55240*	PF05042 (63–228)	-
*BnaOBMP.SH1*	*BnaA01G0146600ZS*	A01:8662825-8663674:+	564	187	20.68	9.45	16.04	80.86	-0.365	*AT4G25140*	PF01277 (56–166)	-
*BnaOBMP.SH2*	*BnaA03G0484200ZS*	A03:26812679-26813659:+	552	183	20.00	9.3	16.11	84.81	−0.234	*AT4G25140*	PF01277 (47–157)	-
*BnaOBMP.SH3*	*BnaA08G0173700ZS*	A08:21101718-21102518:−	543	180	19.51	9.15	20.39	84.61	−0.248	*AT4G25140*	PF01277 (46–156)	-
*BnaOBMP.SH4*	*BnaA10G0084300ZS*	A10:13352481-13353065:+	450	149	15.57	10.1	30.73	104.09	0.286	*AT5G51210*	PF01277 (30–133)	-
*BnaOBMP.SH5*	*BnaC01G0186400ZS*	C01:13785999-13786868:+	564	187	20.69	9.33	13.84	77.22	−0.392	*AT4G25140*	PF01277 (56–166)	-
*BnaOBMP.SH6*	*BnaC03G0340600ZS*	C03:23254070-23254745:+	582	193	20.74	8.11	33.2	97.46	0	*AT3G01570*	PF01277 (35–146)	-
*BnaOBMP.SH7*	*BnaC07G0462300ZS*	C07:55880971-55881762:+	561	186	20.56	9.15	18.99	82.9	−0.354	*AT4G25140*	PF01277 (47–156)	-
*BnaOBMP.SH8*	*BnaC08G0164400ZS*	C08:27380598-27381601:+	537	178	19.34	9.15	23.34	85.56	−0.266	*AT4G25140*	PF01277 (46–156)	-
*BnaOBMP.SH9*	*BnaC09G0330400ZS*	C09:41213467-41214734:+	450	149	15.71	9.99	27.21	104.09	0.311	*AT5G51210*	PF01277 (30–129)	-
*BnaOBMP.SL1*	*BnaA02G0345800ZS*	A02:30864089-30865051:+	567	188	20.00	9.15	30.63	97.02	−0.005	*AT3G27660*	PF01277 (47–159)	-
*BnaOBMP.SL2*	*BnaA04G0117500ZS*	A04:13444417-13445612:−	663	220	23.02	9.1	37.94	79.36	−0.051	*AT5G40420*	PF01277 (69–181)	N
*BnaOBMP.SL3*	*BnaA07G0173200ZS*	A07:19858586-19859760:−	633	210	22.07	9.55	32.57	82.71	−0.085	*AT5G40420*	PF01277 (62–175)	N
*BnaOBMP.SL4*	*BnaA09G0039900ZS*	A09:2460103-2460989:+	567	188	19.87	6.91	37.42	96.97	0.045	*AT3G27660*	PF01277 (47–159)	-
*BnaOBMP.SL5*	*BnaC02G0465400ZS*	C02:57176242-57177199:+	576	191	20.24	7.89	25.31	101.1	0.058	*AT3G27660*	PF01277 (49–161)	-
*BnaOBMP.SL6*	*BnaC04G0404100ZS*	C04:52895656-52896822:−	663	220	23.08	8.78	38.78	76.73	−0.093	*AT5G40420*	PF01277 (69–181)	N
*BnaOBMP.SL7*	*BnaC06G0165300ZS*	C06:26572577-26573813:−	633	210	22.12	9.42	30.96	84.1	−0.102	*AT5G40420*	PF01277 (62–175)	N
*BnaOBMP.SL8*	*BnaC09G0025400ZS*	C09:1687415-1688286:+	567	188	19.84	6.97	34.88	96.44	0.048	*AT3G27660*	PF01277 (47–159)	-
*BnaOBMP.T1*	*BnaA02G0026000ZS*	A02:1608761-1609763:−	609	202	19.68	9.6	44.16	107.62	0.9	*AT5G07600*	PF01277 (11–90)	-
*BnaOBMP.T2*	*BnaA02G0026100ZS*	A02:1624573-1625565:−	498	165	15.93	11	51.09	107.58	0.704	*AT5G07571*	PF01277 (17–102)	N; M
*BnaOBMP.T3*	*BnaA02G0026200ZS*	A02:1626996-1628100:−	450	149	14.90	11.56	55.55	108.26	0.632	*AT5G07560*	PF01277 (20–104)	N; M
*BnaOBMP.T4*	*BnaA03G0028000ZS*	A03:1342717-1343698:−	801	266	26.61	10.21	56.37	78.05	0.056	*AT5G07530*	PF01277 (11–94)	N
*BnaOBMP.T5*	*BnaA03G0028100ZS*	A03:1345527-1346506:−	582	193	20.31	10.16	22.95	88.45	−0.041	*AT5G07530*	PF01277 (29–138)	N
*BnaOBMP.T6*	*BnaA03G0028200ZS*	A03:1348599-1348838:−	240	79	7.93	5.75	23.08	136.08	1.597	*AT5G07550*	PF01277 (6–79)	-
*BnaOBMP.T7*	*BnaA03G0028300ZS*	A03:1357612-1358652:−	354	117	12.12	10.3	24.94	126.75	0.907	*AT5G07571*	PF01277 (20–102)	-
*BnaOBMP.T8*	*BnaA03G0411600ZS*	A03:22258477-22262582:+	798	265	29.45	8.98	52.66	93.09	−0.191	*AT5G61610*	PF01277 (15–101)	-
*BnaOBMP.T9*	*BnaA08G0259200ZS*	A08:25566988-25567994:−	630	209	20.34	9.7	40.66	99.43	0.75	*AT5G07520*	PF01277 (13–93)	-
*BnaOBMP.T10*	*BnaA10G0262800ZS*	A10:24709253-24710147:−	528	175	17.02	10.12	36.57	111.09	0.85	*AT5G07520*	PF01277 (11–92)	-
*BnaOBMP.T11*	*BnaA10G0262900ZS*	A10:24715581-24716318:−	588	195	19.56	10.91	48.78	119.33	0.832	*AT5G07530*	PF01277 (32–112)	-
*BnaOBMP.T12*	*BnaA10G0263000ZS*	A10:24721512-24722645:−	1134	377	37.64	9.46	31.54	50.77	−0.699	*AT5G07530*	PF01277 (34–91)	N
*BnaOBMP.T13*	*BnaA10G0263100ZS*	A10:24729750-24731926:−	1041	346	35.13	9.51	22.74	52.6	−0.846	*AT5G07540*	PF01277 (10–95)	N
*BnaOBMP.T14*	*BnaA10G0263200ZS*	A10:24734513-24735216:−	327	108	11.03	11.17	23.9	116.57	0.807	*AT5G07550*	PF01277 (4–87)	N
*BnaOBMP.T15*	*BnaC02G0028100ZS*	C02:1962414-1963372:−	564	187	18.61	9.52	34.62	109.84	0.86	*AT5G07600*	PF01277 (11–90)	-
*BnaOBMP.T16*	*BnaC02G0028400ZS*	C02:1980601-1981530:−	474	157	15.23	10.72	49.53	109.87	0.73	*AT5G07571*	PF01277 (18–102)	N; M
*BnaOBMP.T17*	*BnaC02G0028500ZS*	C02:1982990-1984077:−	450	149	14.82	11.28	50.09	107.65	0.682	*AT5G07560*	PF01277 (20–104)	N; M
*BnaOBMP.T18*	*BnaC03G0034800ZS*	C03:1785288-1786673:−	630	209	21.99	10.13	25.99	81.67	−0.219	*AT5G07530*	PF01277 (29–138)	N
*BnaOBMP.T19*	*BnaC03G0034900ZS*	C03:1800558-1801344:−	351	116	12.05	9.3	20.83	106.12	0.459	*AT5G07550*	PF01277 (7–89)	N
*BnaOBMP.T20*	*BnaC03G0035100ZS*	C03:1811774-1812971:−	354	117	12.08	10.3	26.11	139.32	1.065	*AT5G07571*	PF01277 (20–100)	-
*BnaOBMP.T21*	*BnaC07G0383700ZS*	C07:50568325-50569786:+	756	251	27.85	9.22	51.45	94.82	−0.139	*AT5G61610*	PF01277 (14–101)	-
*BnaOBMP.T22*	*BnaC08G0242800ZS*	C08:33696624-33697687:+	699	232	21.83	9.7	48.59	92.59	0.678	*AT5G07520*	PF01277 (13–92)	-
*BnaOBMP.T23*	*BnaC09G0577200ZS*	C09:65491385-65492265:−	516	171	16.66	10.39	38.55	107.37	0.839	*AT5G07520*	PF01277 (11–92)	-
*BnaOBMP.T24*	*BnaC09G0577300ZS*	C09:65495132-65495850:−	561	186	18.82	10.91	48.1	121.34	0.844	*AT5G07530*	PF01277 (32–112)	-
*BnaOBMP.T25*	*BnaC09G0577400ZS*	C09:65498951-65500225:−	1275	424	42.08	9.52	36.75	48.8	−0.714	*AT5G07530*	PF01277 (34–94)	N
*BnaOBMP.T26*	*BnaC09G0577500ZS*	C09:65503871-65505231:−	1038	345	34.84	9.47	31.35	53.88	−0.827	*AT5G07540*	PF01277 (10–95)	N
*BnaOBMP.T27*	*BnaC09G0577600ZS*	C09:65506819-65507518:−	327	108	11.06	10.64	27.25	112.96	0.776	*AT5G07550*	PF01277 (4–87)	N
*BnaOBMP.U1*	*BnaA01G0326200ZS*	A01:30207288-30207782:+	495	164	18.14	9.39	32.91	96.4	0.232	*AT3G18570*	PF01277 (43–150)	-
*BnaOBMP.U2*	*BnaA03G0354100ZS*	A03:18830098-18830595:−	498	165	17.82	6.38	40.14	95.76	0.421	*AT3G18570*	PF01277 (45–151)	-
*BnaOBMP.U3*	*BnaA04G0171900ZS*	A04:17911089-17911529:+	441	146	15.65	8.66	40.03	92.88	0.277	*AT2G25890*	PF01277 (27–135)	-
*BnaOBMP.U4*	*BnaA05G0373400ZS*	A05:37373072-37373569:+	498	165	17.93	9.52	33.39	97.52	0.339	*AT3G18570*	PF01277 (43–151)	-
*BnaOBMP.U5*	*BnaC01G0403700ZS*	C01:46777791-46778285:+	495	164	17.86	9.1	28.25	98.17	0.347	*AT3G18570*	PF01277 (43–150)	-
*BnaOBMP.U6*	*BnaC03G0429600ZS*	C03:29754802-29755299:−	498	165	17.77	6.9	37.83	101.7	0.474	*AT3G18570*	PF01277 (45–151)	-
*BnaOBMP.U7*	*BnaC04G0469500ZS*	C04:59628156-59628593:+	438	145	15.51	7.85	38.17	94.21	0.308	*AT2G25890*	PF01277 (26–134)	-
*BnaOBMP.U8*	*BnaC05G0413200ZS*	C05:46507388-46507885:+	498	165	17.99	9.52	32.87	96.3	0.352	*AT3G18570*	PF01277 (43–151)	-
*BnaOBMP.U9*	*BnaC08G0042100ZS*	C08:3813516-3813938:−	423	140	15.61	10.16	63.2	108.71	0.513	*AT1G48990*	PF01277 (48–140)	-
*BnaOBMP.S1*	*BnaA01G0228500ZS*	A01:15029365-15042028:−	651	216	24.50	5.86	24.88	114.63	0.381	*AT3G47360*	PF00106 (48–144);	N
*BnaOBMP.S2*	*BnaA02G0152000ZS*	A02:8745424-8747467:+	936	311	34.33	8.04	28.99	103.44	0.342	*AT5G50770*	PF00106 (48–237);	-
*BnaOBMP.S3*	*BnaA02G0152200ZS*	A02:8762187-8763861:+	1050	349	39.04	6.04	39.72	90.77	0.052	*AT5G50700*	PF00106 (48–236);	-
*BnaOBMP.S4*	*BnaA03G0246000ZS*	A03:12864562-12866180:+	1026	341	38.28	6.16	34.07	92.61	0.119	*AT5G50700*	PF00106 (48–235);	-
*BnaOBMP.S5*	*BnaA03G0253100ZS*	A03:13272696-13274906:+	1386	461	51.61	6.84	61.57	73.99	−0.305	*AT4G10020*	PF00106 (50–237);	N
*BnaOBMP.S6*	*BnaA03G0430100ZS*	A03:23315422-23317643:−	930	309	34.69	8.77	27.53	110.1	0.302	*AT3G47360*	PF00106 (48–236);	N
*BnaOBMP.S7*	*BnaA06G0259900ZS*	A06:36071053-36075029:+	933	310	35.30	7.64	24.24	108.81	0.214	*AT3G47360*	PF00106 (48–236);	-
*BnaOBMP.S8*	*BnaA10G0081400ZS*	A10:12976837-12979574:+	900	299	33.14	8.91	42.98	108.23	0.233	*AT5G50690*	PF00106 (48–234);	N; M
*BnaOBMP.S9*	*BnaC01G0292700ZS*	C01:26062383-26082083:−	564	187	20.93	6.15	22.08	100.11	0.16	*AT3G47360*	PF00106 (11–161);	N
*BnaOBMP.S10*	*BnaC02G0194700ZS*	C02:16258391-16263882:+	945	314	34.79	8.86	27.39	105.25	0.285	*AT5G50770*	PF00106 (48–244);	-
*BnaOBMP.S11*	*BnaC02G0194900ZS*	C02:16283980-16285900:+	1050	349	39.11	6.26	40.91	89.94	0.046	*AT5G50700*	PF00106 (48–236);	-
*BnaOBMP.S12*	*BnaC03G0290800ZS*	C03:18818616-18820197:+	1026	341	38.24	6.16	33.39	92.02	0.109	*AT5G50700*	PF00106 (48–235);	-
*BnaOBMP.S13*	*BnaC03G0300500ZS*	C03:19724445-19726858:+	1392	463	51.97	6.71	66.84	73.46	−0.331	*AT4G10020*	PF00106 (50–237);	N
*BnaOBMP.S14*	*BnaC07G0404100ZS*	C07:51850357-51852617:+	930	309	34.73	8.77	25.47	110.42	0.32	*AT3G47360*	PF00106 (48–236);	N
*BnaOBMP.S15*	*BnaC09G0327500ZS*	C09:40510021-40511538:−	900	299	33.26	9.05	45.99	108.23	0.225	*AT5G50690*	PF00106 (48–235);	N; M

^1^ Chromosome: start position–end position: strand; (−) antisense strand of chromosome; (+) positive-sense strand of chromosome; ^2^ nucleus (N); mitochondria (M).

## Data Availability

Not applicable.

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
