# Peer review of "Genome-Wide Identification and Characterization of Oil-Body-Membrane Proteins in Polyploid Crop Brassica napus"

_plants, 2022, doi:10.3390/plants11172241_

Round 1

Reviewer 1 Report (Previous Reviewer 1)

The authors have carefully answered my comments and concern in the revised manuscript, and thus it is recommended for publication.

Author Response

Thanks.

Reviewer 2 Report (Previous Reviewer 2)

Congratulation!

The improvements are clearly and increase the quality and significance  of the content.

Author Response

Thanks.

Reviewer 3 Report (Previous Reviewer 3)

The authors responded to my comments and amendments were introduced. The nomenclature for gene and protein is still not improved. 

Author Response

We did not see this comment in the previous peer review comments. However, we still checked and revised the nomenclature for gene and protein in the manuscript (lines 386-392 marked up by the “Track Changes”).

This manuscript is a resubmission of an earlier submission. The following is a list of the peer review reports and author responses from that submission.

Round 1

Reviewer 1 Report

 In this study, the authors performed a genome-wide identification of family genes encoding oil-body membrane proteins (OBMPs) in rapeseed Brassica napus. Gene structures, phylogenetic relationships, cis-acting regulatory elements in the promoter regions, sequence polymorphisms, and expression patterns of the identified OBMPs were theoretically analyzed.  Based on the theoretical analyses, the authors claim that these findings lay foundations for further elucidating molecular functions of Brassica napus OBMPs and provide potential important targets for rapeseed breeding to increase seed oil content.

1.          This work provides documentation of tedious sequence information. Unfortunately, the authors failed to point out novel information revealed by their analyses of data. Counting how many genes (numbers) of oleosin, caleosin and steroleosin as well as indicating what types of putative cis-acting regulatory elements in the promoter regions does not lead to the advancement of this research area scientifically. The hardworking of data accumulation is appreciated, but the authors should pay more attention on the scientific significance originated from their sequence analysis.  

2.          In the analysis, the authors found that 12 Brassica napus OBMP genes harbored strong selection signatures, likely having experienced selection during rapeseed domestication and breeding. This seems to be an interesting topic. The authors are suggested to provide more evidence to support this hypothesis.

3.          In the end, the authors conclude that these findings lay foundations for further elucidating molecular functions of Brassica napus OBMPs and provide potential important targets for rapeseed breeding to increase seed oil content.  It is not clear how to approach this achievement. Please describe or propose some workable approaches to increase seed oil content of rapeseed breeding according to the findings from this study.

Reviewer 2 Report

Congratulations for your team work! The paper provides for further elucidating the functional roles of OBMP family genes and potential targets for engineering to improve SOC in B. napus The authors publish results of hard and intensive work but the paper have to be improved in some paragraphs. In my document there was no chapter of conclusion (at last draft sending) and no discussions (at the beginning draft) . In my opinion you have to underline your novel information revealed by your research and write more or the key points of your research which can be used in order to increase seed oil content. The last, but not the list, please write clear conclusions in a specific ending paragraph!

Reviewer 3 Report

All work is based on bioinformatic estimations. There is no confirmation in the experimental data. Typically, RNA-seq data is confirmed by qPCR and variants of variation are confirmed by PCR and sequencing. This is a fundamental issue, confirming the significance of the analyses, which significantly affects the evaluation of the paper. It is necessary to confirm the data with experimental validation.

There are no conclusions in the paper.